# Air Pollution Is Associated with Poor Cognitive Function in Taiwanese Adults

**DOI:** 10.3390/ijerph18010316

**Published:** 2021-01-04

**Authors:** Meng-Chieh Chen, Chen-Feng Wang, Bo-Cheng Lai, Sun-Wung Hsieh, Szu-Chia Chen, Chih-Hsing Hung, Chao-Hung Kuo

**Affiliations:** 1Department of General Medicine, Kaohsiung Medical University Hospital, Kaohsiung 807, Taiwan; b0103122@gmail.com; 2Institute of Electronics, National Chiao Tung University, Hsinchu 300, Taiwan; a114n.d425y@gmail.com (C.-F.W.); bclai@mail.nctu.edu.tw (B.-C.L.); 3Department of Neurology, Kaohsiung Medical University Hospital, Kaohsiung Medical University, Kaohsiung 807, Taiwan; circle.6@yahoo.com.tw; 4Department of Neurology, Kaohsiung Municipal Siaogang Hospital, Kaohsiung Medical University, Kaohsiung 812, Taiwan; 5Neuroscience Research Center, Kaohsiung Medical University, Kaohsiung 807, Taiwan; 6Division of Nephrology, Department of Internal Medicine, Kaohsiung Medical University Hospital, Kaohsiung Medical University, Kaohsiung 807, Taiwan; 7Department of Internal Medicine, Kaohsiung Municipal Siaogang Hospital, Kaohsiung Medical University, Shan-Ming Rd., Hsiao-Kang Dist., Kaohsiung 812, Taiwan; kjh88kmu@gmail.com; 8Faculty of Medicine, College of Medicine, Kaohsiung Medical University, Kaohsiung 807, Taiwan; 9Research Center for Environmental Medicine, Kaohsiung Medical University, Kaohsiung 807, Taiwan; pedhung@gmail.com; 10Department of Pediatrics, Kaohsiung Medical University Hospital, Kaohsiung Medical University, Kaohsiung 807, Taiwan; 11Department of Pediatrics, Kaohsiung Municipal Siaogang Hospital, Kaohsiung Medical University, Kaohsiung 812, Taiwan; 12Division of Gastroenterology, Department of Internal Medicine, Kaohsiung Medical University Hospital, Kaohsiung Medical University, Kaohsiung 807, Taiwan

**Keywords:** air pollutants, cognitive decline, mini-mental state exam, sub-domains

## Abstract

The issue of air pollution is gaining increasing attention worldwide, and mounting evidence has shown an association between air pollution and cognitive decline. The aim of this study was to investigate the relationships between air pollutants and cognitive impairment using the Mini-Mental State Exam (MMSE) and its sub-domains. In this study, we used data from the Taiwan Biobank combined with detailed daily data on air pollution. Cognitive function was assessed using the MMSE and its five subgroups of cognitive functioning. After multivariable linear regression analysis, a high level of particulate matter with an aerodynamic diameter of ≤2.5 μm (PM_2.5_), low ozone (O_3_), high carbon monoxide (CO), high sulfur dioxide (SO_2_), high nitric oxide (NO), high nitrogen dioxide (NO_2_), and high nitrogen oxide (NO_x_) were significantly associated with low total MMSE scores. Further, high SO_2_ and low O_3_ were significantly associated with low MMSE G1 scores. Low O_3_, high CO, high SO_2_, high NO_2_, and high NO_x_ were significantly associated with low MMSE G4 scores, and high PM_2.5_, high particulate matter with an aerodynamic diameter of ≤10 μm (PM_10_), high SO_2_, high NO_2_, and high NO_x_ were significantly associated with low MMSE G5 scores. Our results showed that exposure to different air pollutants may lead to general cognitive decline and impairment of specific domains of cognitive functioning, and O_3_ may be a protective factor. These findings may be helpful in the development of policies regarding the regulation of air pollution.

## 1. Introduction

Air pollution has become a serious concern worldwide. The World Health Organization (WHO) estimates that only one in ten people breathe air that does not contain pollutants exceeding the recommended limits. In the Asia-Pacific region, including Taiwan, air pollution is substantially more severe than in most developed countries, mostly due to rapid urbanization and industrialization with fewer emission regulations [1]. Air pollution has been shown to pose threats to health, and it has been linked to many diseases, including cardiovascular diseases, chronic obstructive pulmonary diseases, and even autoimmune diseases [2]. Emerging evidence has also shown associations between air pollution and cognitive decline and neurological disorders, such as Alzheimer’s disease and Parkinson’s disease [3], which will increase the economic burden on our aging society. Thus, there is a growing need to identify the relationships between air pollution and cognitive decline.

Various tools are available to measure cognitive function, such as the General Practitioner Assessment of Cognition, Mini-Cog™, and Mini-Mental State Exam (MMSE). Of these, the MMSE is a widely accepted tool that is easy to administer and divided into five subgroups to evaluate different aspects of cognitive functioning [4]. Most previous epidemiological studies investigating air pollution and cognitive impairment have focused on a limited number of pollutants, such as particulate matter (PM) and ozone (O_3_). Experimental research has demonstrated that PM exposure can lead to diffuse accumulation of cerebral beta amyloid plaque, hyperphosphorylated tau pre-tangles, and peripheral systemic inflammation, which further result in the activation of microglial and astrocytes in the central nervous system (CNS). Moreover, neuroinflammation has been implicated as an important pathway in neurodegenerative disease [5]. O_3_, per se, is a strong oxidative pollutant, which elicits neurological injury by induction of release of free radicals, activation the generation of inflammatory cytokines and compromising the integrity of the blood–brain barrier. Despite most evidence showing its detrimental influence on cognition, a few epidemiological studies revealed protective effects at low concentrations of O_3_ exposure [6]. However, on the other hand, few studies have investigated other air pollutants, such as carbon monoxide (CO), sulfur dioxide (SO_2_), nitric oxide (NO), nitrogen dioxide (NO_2_), and nitrogen oxide (NO_x_) [7,8], and the mechanisms on cognitive decline are still vague. Moreover, the results of relationships between these air pollutants and global cognitive function, and different trajectories of cognitive functioning, have been inconsistent [5].

In this study, we combined data from two databases, the Taiwan Biobank (TWB) and Taiwan Air Quality Monitoring Database (TAQMD). The TWB included 5000 registered individuals, and the TAQMD was comprised of detailed daily data on air pollution in Taiwan. Choe YM et al. evaluated the usefulness of subscores on the MMSE for predicting the progression of Alzheimer’s disease dementia in individuals with mild cognitive impairment. They found that MMSE memory, orientation, and construction subscores, which are simple and readily available clinical measures, could provide useful information to predict Alzheimer’s disease dementia progression in individuals with mild cognitive impairment, in practical clinical settings [9]. Therefore, we are very interested in knowing the effect of each air pollutant on its five subgroups of cognitive functioning. Thus, the aim of this study was to explore associations between the most well-known atmospheric air pollutants and cognitive impairment using the MMSE and its subdomains.

## 2. Materials and Methods

### 2.1. Ethics Statement

The Institutional Review Board on Biomedical Science Research, Academia Sinica, Taiwan, and the Ethics and Governance Council of the TWB, Taiwan granted ethical approval to the TWB. Each participant provided written informed consent, and the study was conducted according to institutional guidelines and the principles of the Declaration of Helsinki. In addition, this study was approved by the Institutional Review Board of Kaohsiung Medical University Hospital (KMUHIRB-E(I)-20180242).

### 2.2. The Taiwan Biobank

The TWB was established to record genomic and lifestyle data of Taiwanese residents, and it is currently the largest government-supported biobank in Taiwan [10,11]. The TWB is comprised of data of community-based volunteers aged 30 to 70 years with no history of cancer. A total of 5000 individuals who were registered in the TWB in April 2014 were enrolled in this study. All of the participants signed informed consent forms, gave blood samples, and underwent physical examinations. Data, including body height and weight, and body mass index (BMI) (kg/m^2^), were recorded. The participants were also asked to complete questionnaires about personal and lifestyle factors in face-to-face interviews with TWB researchers.

### 2.3. Collection of Demographic, Medical and Laboratory Data

Demographic data (age and sex), smoking history, medical history (diabetes mellitus (DM) and hypertension), examination findings (systolic (SBP) and diastolic blood pressures (DBP)) and laboratory data (fasting glucose, triglycerides, total cholesterol, hemoglobin, estimated glomerular filtration rate (eGFR) and uric acid) were recorded at baseline. EGFR was calculated using the four-variable Modification of Diet in Renal Disease equation [12].

### 2.4. Evaluation of Cognitive Function

We assessed the cognitive function of the subjects using the MMSE [4]. The participants completed MMSE questionnaires in face-to-face interviews with TWB researchers on the day of the enrollment. The MMSE is used as a screening tool for cognitive impairment, in which lower scores indicate that further evaluations are warranted. The MMSE contains five subscales: G1 orientation (score 0–10) (orientation to time and orientation to place), G2 registration (score 0–3), G3 attention and calculation (score 0–3), G4 recall (score 0–3), and G5 language (score 0–11) (including reading, repeat, naming, sentence, construction and obey). Total MMSE scores were calculated as the sum of each sub-score, with a maximum score of 30. Participants older than 60 years were invited to take the MMSE, and a total of 1054 participants who completed the MMSE questionnaire during the enrollment period were included in this study.

### 2.5. Assessment of Air Pollutants

The TAQMD was established by the Taiwan Environmental Protection Administration, Executive Yuan, and includes data on daily air pollutant concentrations at 74 ambient air quality monitoring stations around Taiwan. In this study, we linked the TAQMD and TWB by the location of the air quality monitoring stations and the area of residence of the participants. The residential address of each participant was used to estimate exposure to outdoor air pollution. The average concentrations of air pollutants, including PM with an aerodynamic diameter of ≤2.5 μm (PM_2.5_), PM with an aerodynamic diameter of ≤10 μm (PM_10_), O_3_, CO, SO_2_, NO, NO_2_, and NO_x_ during each year were calculated, and average yearly data were determined in three steps: (1) the corresponding longitude and latitude of the address were obtained using Google geocoding; (2) the nearest air quality monitoring station was identified from an interpolation point; (3) data from the this station were filtered from the survey date to the previous year, and the average of each air pollution metric was calculated.

### 2.6. Example of Nearest Neighbor Interpolation

To illustrate our interpolation method and dataset distribution, we plotted the locations of the monitoring stations and participants as shown in the previous article [13].

### 2.7. Statistical Analysis

Data were expressed as mean ± standard deviation, frequency (%), or median (25th–75th percentile) for triglycerides. An MMSE cut-off score of 24 was used to classify the severity of cognitive impairment. Differences between groups were analyzed using the chi-square test for categorical variables and the independent t test for continuous variables. Linear regression analysis was used to identify associations between each air pollutant and MMSE and subscales. A *p* value of less than 0.05 was considered to indicate a statistically significant difference. Statistical analysis was performed using SPSS version 19.0 for Windows (SPSS Inc. Chicago, IL, USA).

## 3. Results

The mean age of the 1054 participants was 64.0 ± 2.9 years, and included 519 males and 535 females. The participants were stratified into two groups according to an MMSE score ≥24 (n = 914, 86.7%) or <24 (n = 140, 13.3%). Comparisons of the clinical characteristics between these two groups are shown in Table 1. Compared to the participants with an MMSE score ≥24, those with an MMSE score < 24 were older, more predominantly female, had lower education years, low O_3_, lower total MMSE score, and lower score on each MMSE subscale.

Table 2 shows the determinants of total MMSE score in the study participants using multivariable linear regression analysis. Old age (unstandardized coefficient β, −0.006; *p* = 0.009), female (unstandardized coefficient β, 0.534; *p* = 0.002), low education years (unstandardized coefficient β, 0.950; *p* = 0.039), high BMI (unstandardized coefficient β, −0.092; *p* = 0.001), low O_3_ (unstandardized coefficient β, 0.047; *p* < 0.029), and high SO_2_ (unstandardized coefficient β, −0.145; *p* = 0.022) were associated with low total MMSE score.

### 3.1. Correlations between Air Pollutants and Total MMSE Scores

Table 3 shows the determinants of total MMSE score in the study participants using multivariable linear regression analysis after adjusting for living alone, being employed, age, sex, smoking and alcohol history, DM, hypertension, cerebrovascular disease, BMI, SBP, DBP, fasting glucose, log triglycerides, total cholesterol, hemoglobin, eGFR, uric acid and each air pollutant. High PM_2.5_ (unstandardized coefficient β, −0.014; *p* = 0.039), low O_3_ (unstandardized coefficient β, 0.078; *p* < 0.001), high CO (unstandardized coefficient β, −1.133; *p* = 0.005), high SO_2_ (unstandardized coefficient β, −0.227; *p* < 0.001), high NO (unstandardized coefficient β, −0.042; *p* = 0.022), high NO_2_ (unstandardized coefficient β, −0.043; *p* < 0.001), and high NO_x_ (unstandardized coefficient β, −0.026; *p* = 0.001) were significantly associated with low total MMSE score, whereas PM_10_ was not significantly associated with total MMSE score.

### 3.2. Correlations between Air Pollutants and Each MMSE Subscore

Table 4 shows the determinants of MMSE G1 (orientation) score in the study participants using multivariable linear regression analysis. The results showed that high SO_2_ (unstandardized coefficient β, −0.044; *p* = 0.015), and low O_3_ (unstandardized coefficient β, 0.014; *p* = 0.020) were significantly associated with a low MMSE G1 score.

Table 5 shows the determinants of MMSE G4 (recall) score in the study participants using multivariable linear regression analysis. The results showed that low O_3_ (unstandardized coefficient β, 0.016; *p* = 0.023), high CO (unstandardized coefficient β, −0.284; *p* = 0.047), high SO_2_ (unstandardized coefficient β, −0.065; *p* = 0.002), high NO_2_ (unstandardized coefficient β, −0.010; *p* = 0.019) and high NO_x_ (unstandardized coefficient β, −0.006; *p* = 0.026) were significantly associated with a low MMSE G4 score.

Table 6 demonstrates the determinants of MMSE G5 (language) score in the study participants using multivariable linear regression analysis. The results showed that high PM_2.5_ (unstandardized coefficient β, −0.005; *p* = 0.022), high PM_10_ (unstandardized coefficient β, −0.003; *p* = 0.013), high SO_2_ (unstandardized coefficient β, −0.057; *p* = 0.003), high NO_2_ (unstandardized coefficient β, −0.011; *p* = 0.006), and high NO_x_ (unstandardized coefficient β, −0.006; *p* = 0.020) were significantly associated with a low MMSE G5 score.

We also analyzed correlations between air pollutants and MMSE G2 (registration) and G3 (attention and calculation) scores. However, no air pollutants were significantly associated with these scores, except for low O_3_ (unstandardized coefficient β, 0.033; *p* = 0.011), which was associated with a low MMSE G3 score.

## 4. Discussion

In this study of 1054 participants registered in the TWB from around Taiwan, we observed that high PM_2.5_, CO, SO_2_, NO, NO_2_, and NO_x_, and low O_3_ were correlated with low total MMSE score. We further found that these pollutants were separately correlated with different MMSE subscores.

The first important finding of this study is that high PM_2.5_ was associated with low total MMSE and MMSE G5 (language) scores, and that high PM_10_ was also related with a low MMSE G5 score. Increasing epidemiological evidence has shown the effect of PM_2.5_ on worsening cognitive function [13,14,15,16]. In one prospective cohort study, Wang et al. reported the adverse effect of long-term exposure to PM_2.5_ on worsening cognitive function [17]. In addition, Calderón-Garcidueñas also reported that exposure to PM_2.5_ could also cause deficits in multiple domains of cognitive function, including executive, memory, visuospatial, and language functions in young adults [18]. The primary mechanism by which PM damages the cardiovascular and pulmonary systems is mainly by inducing both local and systemic oxidative stress and inflammation [19]; however, the detailed mechanism by which PM damages the CNS has yet to be elucidated. Once inhaled, PM can cause direct cell damage via free radicals and stimulate the release of inflammatory cytokines, which can then induce the release of cytokines and reactive oxygen species in the brain. This then leads to a cycle of neuroinflammation and oxidative damage, and finally injury to the CNS [20,21,22]. PM can also make the CNS more susceptible to systemic inflammation by destroying the integrity of the blood–brain barrier (BBB) after directly entering the brain via the olfactory nerve and altering microbial balance in the gastrointestinal tract, which indirectly leads to an imbalance of immunity in the CNS [23]. With regards to pathological changes in the brain, growing evidence supports that airborne PM_2.5_ accelerates the accumulation of cerebral β-amyloid and tau hyper-phosphorylation [24,25], which are typical in the development of Alzheimer disease. From a macroscopic perspective, a previous study demonstrated an association between PM_2.5_ and smaller hippocampal volume [26]. Taken together, these findings indicate that exposure to PM_2.5_ is a risk for the development of cognitive decline and CNS damage. However, results of the association between cognitive decline and the coarser particles of PM_10_, with an aerodynamic diameter of 2.5 to 10 mm, have been inconsistent [27,28,29,30]. Shin et al. proposed that the difference between PM_2.5_ and PM_10_ is the ability to more readily cross the BBB and cause more systematic inflammation [16], although numerous studies have suggested that nanoparticles, some of which are larger than PM_10_, can reach the CNS via numerous routes, including inhalation through the lungs, direct capture by the olfactory nerve, digestion into the gastrointestinal tract, or cutaneous pathways [22]. These findings may imply that exposure to PM pollutants can lead to cognitive decline.

The second important finding of this study is that high O_3_ was associated with high total MMSE score and G1, G3, G4 subscores, implying the possible neuroprotective effect of O_3_ on cognitive function. Previous experimental and epidemiological studies have identified that O_3_, a strong oxidative pollutant and, therefore, potentially toxic to humans, has detrimental effects on the CNS [13,14,18,30]. Synergistic and dose effects with PM have also been reported [13,31]. Inhaled O_3_ not only directly damages lung function, but also cause systemic effects via the generation of free radicals. Oxidative stress then further increases the expression of inducible NO and subsequent cellular response, which leads to increases in leukotrienes, prostaglandins, tumor necrosis factor, interleukin-1, interleukin-6, and interleukin-8. Through the active transport of endothelial cells adjacent to the BBB or via other proposed methods of entry into the CNS, these molecules can affect astrocytes and then compromise the integrity of the BBB, and this process can be accelerated by free radicals caused by exposure to O_3_. This vicious cycle has been suggested to lead to the development of neurological disorders and cognitive impairment [32]. However, in a study in Korea, Shin et al. found a positive association between O_3_ and a variety of measures for cognition including MMSE Korea child scores, digit-forward span scores, word list recall, recall storage, and frontal assessment battery scores [16]. Similar results were also found in a cross-sectional study by Gatto et al. in Los Angeles [30] and a cohort study by Carey et al. in London [33]. The discrepancy may be due to a dose–effect, as the positive findings were linked to similar low concentrations of O_3_ exposure. Although exposure to toxic levels of O_3_ as an air pollutant has been considered to induce systemic inflammation and damage, exposure to low O_3_ concentrations has been used for medical purposes via activating Nrf2, which can enhance the activity of antioxidant enzymes such as superoxide dismutase and catalase [34]. This may provide a mechanism by which O_3_ has a positive effect on the CNS in contrast to the other reported detrimental effects. Further experimental and epidemiological studies are needed to investigate the relationships between O_3_ as an air pollutant and the CNS to fill this gap in knowledge.

The third important finding of this study is that high CO was correlated with low total MMSE score and recall domain scores, suggesting the detrimental effect on cognitive function. Chang et al. demonstrated the negative effect of CO on cognitive performance [6], and Shin et al. also reported associations between increased risks of poor global cognition, low attention, and executive function and CO exposure [16]. CO has a 200-fold greater affinity for hemoglobin than oxygen, and the decreased binding capacity of oxygen can lead to a reduction in the amount of oxygen reaching the periphery tissue, including the brain. Acute exposure to CO can cause tissue hypoxia-ischemia, cardiac compromise, and shock, which can further worsen hypoxia of the CNS and subsequently cause neurological damage [35,36]. Nakamura et al. reported possible causality between chronic exposure to CO and cognitive dysfunction in two case reports [37]. In addition, Chen et al. used diffusion tensor imaging to demonstrate that chronic CO intoxication may lead to microstructural damage in corpus callosum sub-regions and cognitive impairment [38]. Previous studies have also suggested that smoking, which exposes people to the toxic constituents of tobacco smoke such as CO, is highly correlated with an increased risk of Alzheimer’s disease. Interestingly, although smoking has been associated with vascular dementia [39], some studies have reported a link between smoking and adverse effects on brain neurobiology and function in people without a history of cardiovascular disease [40]. This may suggest that CO plays a role in cognitive decline.

In this study, we found negative associations between SO_2_ and global cognitive decline and impairment of specific functioning domains including orientation, recall, and language, suggesting its adverse influence on neurocognitive performance. A few studies have investigated the relationships between atmospheric SO_2_ and neurological dysfunction [16,41]. Several studies have reported the effect of SO_2_ on the cardiovascular system [42,43], however few studies have reported the effect on the CNS. Hippocampal synaptic dysfunction due to SO_2_ exposure has been reported, and this may play a role in cognitive impairment [44]. Furthermore, co-exposure to low doses of PM_2.5_ and SO_2_ has been reported to result in neurodegeneration, including neuronal apoptosis, and decreased synaptic structural and functional proteins [45]. However, in the present study, SO_2_ was the one air pollutant related to most impaired trajectories of cognitive functioning according to MMSE subscales, indicating that further studies are warranted to investigate its potential damage on the CNS.

An additional important finding of this study is that high NO, NO_2_, and NO_x_ were associated with low total MMSE scores, and we further found that NO_2_ and NO_x_ were associated with low MMSE G4 and G5 subscores. The neurotoxicity of NO as an air pollutant remains unclear. The physiological regulatory functions of endogenous NO include the modulation of blood vessel tone, immune response to neurotransmission and synaptic plasticity, and an excess of NO has been proven to be toxic [46]. In addition, the overproduction of NO can elicit cellular damage by causing nitrosative stress through the formation of reactive nitrogen species, and the combination of reactive nitrogen species and NO has been shown to be involved in the pathogenesis of neurodegenerative disorders such as Alzheimer’s disease [47,48]. NO can also activate cyclooxygenase (COX), which is increased in the CNS under pro-inflammatory conditions. During its catalytic cycle, COX produces free radicals and prostaglandins, both of which can cause neurotoxicity [49,50]. Many studies have identified that NO_2_ can negatively affect cognitive function in different periods of life [5,7,16]. During the prenatal period, Lertxundi et al. reported that exposure to a 1 μg/m^3^ increase in NO_2_ was associated with a -0.29 point decrease in mental score in two-year-old children, which may suggest its adverse effect on cognitive development [51]. NO_2_ was also associated with an increased incidence of dementia in a cohort study of participants aged from 55 to 85 years [16]. As NO_2_ is a reactive oxidative species, the mechanism by which it damages the CNS can be assumed to involve oxidative stress, to which the brain is especially susceptible [52]. A previous animal study showed that acute NO_2_ exposure could induce morphological changes of the mitochondria in the cortices of rats and interfere with mitochondrial metabolism including increased generation of reactive oxygen species, which in turn affected mitochondrial energy production and biogenesis [53]. These studies may support the hypothesis that exposure to nitric compound pollutants can lead to cognitive decline.

There are several limitations to this study. First, we believe that the relationship between air pollutants and cognitive decline takes a period of time, not a short period of time. However, it is a cross-sectional study, and, therefore, we could not examine the effects of long-term exposure to these air pollutants on cognitive functioning. Follow-up studies are needed to confirm our results. Second, cognitive performance was assessed using only the MMSE and its subscales in our study. MMSE is a screening tool, and it only provides information about cognitive status. As a screening test, it is validated using its total score. Although different items use different cognitive functions, and some of them are summed in subscores, they are not actually sub-scales without psychometric information. Finally, levels of air pollutants were estimated at the home addresses of the participants, which is a crude measure of exposure to air pollution. Although outdoor air pollution is known to substantially contribute to indoor/personal exposure, it may not accurately reflect actual personal exposure; however, we did not have additional information about the air quality indoors.

In conclusion, the results of this study showed that exposure to different air pollutants may result in general cognitive decline and impairment of specific domains of cognitive functioning. In addition, we found that most known air pollutants are associated with cognitive decline, and this may be helpful for the government in the development of policies for early diagnosis and long-term follow-up of neurological deficits of people residing in polluted areas.

## Figures and Tables

**Table 1 ijerph-18-00316-t001:** Comparison of clinical characteristics among participants according to total MMSE scores ≥24 or <24.

Characteristics	All (n = 1054)	MMSE ≥ 24 (n = 914)	MMSE < 24 (n = 140)	*p*
Age (year)	64.0 ± 2.9	63.9 ± 2.9	64.6 ± 2.8	0.015
Male gender (%)	49.2	50.8	39.3	0.011
Smoking history (%)	24.8	25.5	20.0	0.161
Alcohol history (%)	5.7	5.8	5.0	0.704
DM (%)	11.5	11.4	12.1	0.792
Hypertension (%)	24.2	24.4	22.9	0.692
Cerebrovascular disease (%)	1.1	1.2	0.7	1.000
Education (years)	4.8 ± 1.2	5.0 ± 1.1	3.7 ± 1.3	<0.001
Living alone (%)	9.2	9.4	7.9	0.554
Having job (%)	25.0	24.2	30.1	0.146
BMI (kg/m^2^)	24.4 ± 3.0	24.3 ± 3.0	24.8 ± 3.1	0.093
SBP (mmHg)	126.1 ± 17.2	125.9 ± 17.1	127.7 ± 17.8	0.251
DBP (mmHg)	72.4 ± 10.6	72.5 ± 10.7	72.0 ± 10.4	0.617
Laboratory parameters				
Fasting glucose (mg/dL)	102.0 ± 22.4	101.6 ± 22.0	104.1 ± 24.6	0.229
Triglyceride (mg/dL)	100 (73–137)	100 (73–136.25)	106.5 (78–138.5)	0.695
Total cholesterol (mg/dL)	201.1 ± 36.5	201.3 ± 36.9	199.9 ± 34.0	0.678
Hemoglobin (g/dL)	14.0 ± 1.4	14.1 ± 1.4	13.9 ± 1.3	0.383
eGFR (mL/min/1.73 m^2^)	88.5 ± 28.3	88.2 ± 28.3	90.6 ± 28.4	0.354
Uric acid (mg/dL)	5.7 ± 1.4	5.7 ± 1.4	5.7 ± 1.4	0.915
Air pollutants				
PM_2.5_ (μg/m^3^)	35.2 ± 11.5	35.0 ± 11.7	36.5 ± 10.3	0.158
PM_10_ (μg/m^3^)	62.9 ± 19.7	62.7 ± 19.9	64.1 ± 18.1	0.436
O_3_ (ppb)	30.9 ± 4.1	31.0 ± 4.1	30.1 ± 3.8	0.012
CO (ppm)	0.45 ± 0.21	0.45 ± 0.21	0.47 ± 0.20	0.278
SO_2_ (ppb)	3.5 ± 1.4	3.5 ± 1.4	3.7 ± 1.2	0.154
NO (ppb)	4.3 ± 4.4	4.3 ± 4.4	4.7 ± 4.5	0.351
NO_2_ (ppb)	14.6 ± 6.8	14.4 ± 6.9	15.5 ± 6.0	0.064
NO_x_ (ppb)	18.9 ± 10.4	18.7 ± 10.5	20.1 ± 9.8	0.131
MMSE				
G1 (Orientation)	9.5 ± 0.8	9.6 ± 0.6	8.6 ± 1.3	<0.001
G2 (Registration)	2.9 ± 0.4	2.9 ± 0.3	2.7 ± 0.6	<0.001
G3 (Attention and Calculation)	3.7 ± 1.7	4.1 ± 1.5	1.4 ± 1.2	<0.001
G4 (Recall)	2.2 ± 0.9	2.3 ± 0.8	1.3 ± 1.1	<0.001
G5 (Language, construction, and obey)	8.4 ± 0.9	8.5 ± 0.7	7.3 ± 1.2	<0.001
MMSE total	26.7 ± 2.8	27.5 ± 1.9	21.3 ± 1.8	<0.001

Abbreviations. MMSE, mini-mental state examination; DM, diabetes mellitus; BMI, body mass index; SBP, systolic blood pressure; DBP, diastolic blood pressure; eGFR, estimated glomerular filtration rate; PM_2.5_, particle with aerodynamic diameter of 2.5 μm or less; PM_10_, particle with aerodynamic diameter of 10 μm or less; O_3_, ozone; CO, carbon monoxide; SO_2_; sulfur dioxide; NO, nitric oxide; NO_2_, nitrogen dioxide; NO_x_, nitrogen oxide.

**Table 2 ijerph-18-00316-t002:** Association of clinical characteristic and air pollutants with total MMSE scores using univariable linear regression analysis.

Characteristics	Univariable	
	Unstandardized Coefficient β (95% CI)	*p*
Age (per 1 year)	−0.006 (−0.011, −0.002)	0.009
Male gender (vs. female)	0.534 (0.194, 0.874)	0.002
Smoking history	0.340 (−0.055, 0.735)	0.092
Alcohol history	0.355 (−0.381, 1.092)	0.344
DM	−0.359 (−0.895, 0.176)	0.188
Hypertension	−0.104 (−0.502, 0.295)	0.610
Cerebrovascular disease	0.137 (−1.473, 1.746)	0.868
Education (per 1 years)	0.950 (0.825, 1.075)	<0.001
Living alone	−0.012 (−0.058, 0.034)	0.619
Having job	−0.325 (−0.716, 0.067)	0.104
BMI (per 1 kg/m^2^)	−0.092 (−0.148, −0.037)	0.001
SBP (per 1 mmHg)	−0.003 (−0.013, 0.007)	0.547
DBP (per 1 mmHg)	0.011 (−0.005, 0.027)	0.192
Laboratory parameters		
Fasting glucose (per 1 mg/dL)	−0.007 (−0.015, 0.001)	0.069
Triglyceride (log per 1 mg/dL)	−0.539 (−1.354, 0.276)	0.195
Total cholesterol (per 1 mg/dL)	0.002 (−0.002, 0.007)	0.352
Hemoglobin (per 1 g/dL)	0.069 (−0.054, 0.193)	0.272
eGFR (per 1 mL/min/1.73 m^2^)	−0.004 (−0.010, 0.002)	0.171
Uric acid (per 1 mg/dL)	−0.025 (−0.147, 0.097)	0.688
Air pollutants		
PM_2.5_ (per 1 μg/m^3^)	−0.010 (−0.025, 0.005)	0.182
PM_10_ (per 1 μg/m^3^)	−0.004 (−0.013, 0.004)	0.328
O_3_ (per 1 ppb)	0.047 (0.005, 0.089)	0.029
CO (per 1 ppm)	−0.380 (−1.222, 0.462)	0.376
SO_2_ (per 1 ppb)	−0.145 (−0.270, −0.021)	0.022
NO (per 1 ppb)	−0.016 (−0.055, 0.023)	0.426
NO_2_ (per 1 ppb)	−0.023 (−0.048, 0.003)	0.080
NO_x_ (per 1 ppb)	−0.013 (−0.029, 0.004)	0.135

Values expressed as unstandardized coefficient β and 95% confidence interval (CI). Abbreviations are the same as in Table 1.

**Table 3 ijerph-18-00316-t003:** Association of air pollutants with total MMSE scores using multivariable linear regression analysis.

Air Pollutants	Multivariable	
	Unstandardized Coefficient β (95% CI)	*p*
PM_2.5_ (per 1 μg/m^3^)	−0.014 (−0.028, 0)	0.039
PM_10_ (per 1 μg/m^3^)	−0.006 (−0.015, 0.002)	0.131
O_3_ (per 1 ppb)	0.078 (0.039, 0.117)	<0.001
CO (per 1 ppm)	−1.133 (−1.915, −0.351)	0.005
SO_2_ (per 1 ppb)	−0.227 (−0.342, −0.111)	<0.001
NO (per 1 ppb)	−0.042 (−0.079, −0.006)	0.022
NO_2_ (per 1 ppb)	−0.043 (−0.067, −0.019)	<0.001
NO_x_ (per 1 ppb)	−0.026 (−0.042, −0.011)	0.001

Values expressed as unstandardized coefficient β and 95% confidence interval (CI). Abbreviations are the same as in Table 1. Multivariable model: adjusted for age, sex, smoking and alcohol history, DM, hypertension, cerebrovascular disease, life style with living alone and having job, BMI, SBP, DBP, fasting glucose, log triglyceride, total cholesterol, hemoglobin, eGFR, uric acid, and each air pollutants.

**Table 4 ijerph-18-00316-t004:** Association of air pollutants with MMSE G1 (orientation) using multivariable linear regression analysis.

Air Pollutants	Multivariable	
	Unstandardized Coefficient β (95% CI)	*p*
PM_2.5_ (per 1 μg/m^3^)	−0.001 (−0.006, 0.003)	0.532
PM_10_ (per 1 μg/m^3^)	0 (−0.003, 0.002)	0.757
O_3_ (per 1 ppb)	0.014 (0.002, 0.026)	0.020
CO (per 1 ppm)	−0.175 (−0.413, 0.063)	0.150
SO_2_ (per 1 ppb)	−0.044 (−0.079, −0.009)	0.015
NO (per 1 ppb)	−0.010 (−0.021, 0.001)	0.082
NO_2_ (per 1 ppb)	−0.004 (−0.012, 0.003)	0.232
NO_x_ (per 1 ppb)	−0.004 (−0.008, 0.001)	0.128

Values expressed as unstandardized coefficient β and 95% confidence interval (CI). Abbreviations are the same as in Table 1. Multivariable model: adjusted for age, sex, smoking and alcohol history, DM, hypertension, cerebrovascular disease, life style with living alone and having job, BMI, SBP, DBP, fasting glucose, log triglyceride, total cholesterol, hemoglobin, eGFR, uric acid, and each air pollutants.

**Table 5 ijerph-18-00316-t005:** Association of air pollutants with MMSE G4 (recall) using multivariable linear regression analysis.

Air Pollutants	Multivariable	
	Unstandardized Coefficient β (95% CI)	*p*
PM_2.5_ (per 1 μg/m^3^)	−0.002 (−0.007, 0.003)	0.375
PM_10_ (per 1 μg/m^3^)	−0.002 (−0.005, 0.001)	0.178
O_3_ (per 1 ppb)	0.016 (0.002, 0.030)	0.023
CO (per 1 ppm)	−0.284 (−0.565, −0.003)	0.047
SO_2_ (per 1 ppb)	−0.065 (−0.107, −0.024)	0.002
NO (per 1 ppb)	−0.010 (−0.023, 0.003)	0.121
NO_2_ (per 1 ppb)	−0.010 (−0.019, −0.002)	0.019
NO_x_ (per 1 ppb)	−0.006 (−0.012, 0)	0.026

Values expressed as unstandardized coefficient β and 95% confidence interval (CI). Abbreviations are the same as in Table 1. Multivariable model: adjusted for age, sex, smoking and alcohol history, DM, hypertension, cerebrovascular disease, life style with living alone and having job, BMI, SBP, DBP, fasting glucose, log triglyceride, total cholesterol, hemoglobin, eGFR, uric acid, and each air pollutants.

**Table 6 ijerph-18-00316-t006:** Association of air pollutants with MMSE G5 (language, construction, and obey) using multivariable linear regression analysis.

Air Pollutants	Multivariable	
	Unstandardized Coefficient β (95% CI)	*p*
PM_2.5_ (per 1 μg/m^3^)	−0.005 (−0.010, 0)	0.022
PM_10_ (per 1 μg/m^3^)	−0.003 (−0.006,0)	0.013
O_3_ (per 1 ppb)	0.012 (0, 0.025)	0.058
CO (per 1 ppm)	−0.247 (−0.504, 0.011)	0.061
SO_2_ (per 1 ppb)	−0.057 (−0.095, −0.019)	0.003
NO (per 1 ppb)	−0.008 (−0.020, 0.004)	0.196
NO_2_ (per 1 ppb)	−0.011 (−0.019, −0.003)	0.006
NO_x_ (per 1 ppb)	−0.006 (−0.011, 0)	0.020

Values expressed as unstandardized coefficient β and 95% confidence interval (CI). Abbreviations are the same as in Table 1. Multivariable model: adjusted for age, sex, smoking and alcohol history, DM, hypertension, cerebrovascular disease, life style with living alone and having job, BMI, SBP, DBP, fasting glucose, log triglyceride, total cholesterol, hemoglobin, eGFR, uric acid, and each air pollutants.

## Data Availability

The data underlying this study is from the Taiwan Biobank. Due to restrictions placed on the data by the Personal Information Protection Act of Taiwan, the minimal data set cannot be made publicly available. Data may be available upon request to interested researchers. Please send data requests to: Szu-Chia Chen, PhD, MD. Division of Nephrology, Department of Internal Medicine, Kaohsiung Medical University Hospital, Kaohsiung Medical University.

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
