# Peer review of "Air Pollution Is Associated with Poor Cognitive Function in Taiwanese Adults"

_ijerph, 2021, doi:10.3390/ijerph18010316_

Round 1

Reviewer 1 Report

This topic points to the very important and interesting issue of relationship between air pollution and cognitive decline. The authors combined data from the Taiwan Biobank and detailed daily data on air pollutants. Some points reside before publication is possible:
The introduction section plays a significant role in any research paper. However, this length of the introduction section is too short. In other words, literature review should be more accurate and extended.

In addition, the innovation of the paper is insufficient. Aside from taking other air pollutants into account, there should be other innovations in the paper.

The paper only considers the negative impact of air pollutants on the elderly group, however, various air pollutants may also affect the cognitive function of other age groups. If author can increase the information of cognitive function of other age groups, the conclusion of the paper will be more comprehensive and complete.

Although the empirical results are analyzed in this paper, but few targeted policy advices were given. Therefore, if the author can make suggestions based on the research conclusions, the paper will be more practical and instructive.

Good luck for improvement!

Author Response

Reviewer 1

This topic points to the very important and interesting issue of relationship between air pollution and cognitive decline. The authors combined data from the Taiwan Biobank and detailed daily data on air pollutants. Some points reside before publication is possible:

  1. The introduction section plays a significant role in any research paper. However, this length of the introduction section is too short. In other words, literature review should be more accurate and extended.
  2. Ans: Thank you for your suggestion. We have added more in the Introduction Paragraph 2.
  • The experimental researches have demonstrated that PM exposure can lead to diffuse accumulation of cerebral beta amyloid plaques, and hyperphosphorylated tau pre-tangles, and peripheral systemic inflammations, which further result in activation of microglial and astrocyte in central nervous system (CNS) and neuroinflammation has been implicated as an important pathway in neurodegenerative disease [5]. O3 per se is a strong oxidative pollutant, which elicits neurological injury by induction of release of free radicals, activation the generation of inflammatory cytokines and compromising the integrity of blood-brain barrier. Despite most evidence showed its detrimental influence on cognition, a few epidemiological studies revealed protective effect at low concentrations of O3 exposure [6]. (Page 4, Line 20 to Page 5, Line 3)
  •  
  1. In addition, the innovation of the paper is insufficient. Aside from taking other air pollutants into account, there should be other innovations in the paper.  
  2. Ans: Thank you for your question. In this study, we combined two dataset from the Taiwan Biobank and detailed daily data on air pollution. Except total MMSE scores, we further survey the effect of each air pollutants on its five subgroups of cognitive functioning (G1-G5), which is different from previous articles. Therefore, we think, the association between air pollution with MMSE sub-scores is the innovation of this article.
  3. The paper only considers the negative impact of air pollutants on the elderly group, however, various air pollutants may also affect the cognitive function of other age groups. If author can increase the information of cognitive function of other age groups, the conclusion of the paper will be more comprehensive and complete.  
  4. Ans: Thank you for your suggestion. However, in Taiwan Biobank research process, participants aged older than 60 years were invited to take the MMSE. Therefore, we don’t have the information about other age groups.
  5. Although the empirical results are analyzed in this paper, but few targeted policy advices were given. Therefore, if the author can make suggestions based on the research conclusions, the paper will be more practical and instructive.
  6. Ans: Thank you for your suggestion. We have added the suggestion in the conclusion.
  • In addition, we found that most known air pollutants are associated with cognitive decline and this may be helpful for the government in the development of policies for early diagnosis and long-term follow-up neurological deficit of people resided in the polluted area. (Page 15, Line 12-15)

Reviewer 2 Report

The current papers combines data from the Taiwan Biobank and the Taiwan Environmental Protection Administration studies to look for links between air pollution and cognitive decline. The main problem of the paper is the use of MMSE scores. MMSE is a screening tool, and it only provides information about cognitive status. As a screening test, it is validated using its total score. Although different items uses different cognitive functions, and some of them are summed in sub-scores, they are not actually sub-scales since we have not psychometric information about them. Accordingly, the only dependent variable to be properly used in a high impact paper is the MMSE total score. 

Although the role of the health variables is controlled, it might be studied in order to confirm a complementary role in cognitive performance. It is hard to imagine an exclusive direct effect of the air pollution in cognition, but a more complex mediation effect. 

Other comments:

  • In the abstract, the sentence "Our results showed that exposure to different air pollutants may lead to general cognitive decline and impairment of specific domains of cognitive functioning, except for ozone" it is not clear. 
  • Introduction must be longer, with a more in deep review of the potential effects of air polution in its different componente on cognitive functioning (as it is developed in the Discussion, but with a bigger emphasis in its theoretical basis).

  • How cognitive assessment and assessment of air pollutants are temporarily linked?

Author Response

Reviewer 2

The current paper combines data from the Taiwan Biobank and the Taiwan Environmental Protection Administration studies to look for links between air pollution and cognitive decline.

  1. The main problem of the paper is the use of MMSE scores. MMSE is a screening tool, and it only provides information about cognitive status. As a screening test, it is validated using its total score. Although different items uses different cognitive functions, and some of them are summed in sub-scores, they are not actually sub-scales since we have not psychometric information about them. Accordingly, the only dependent variable to be properly used in a high impact paper is the MMSE total score. 
  2. Ans: Thank you for your question. I totally agree your point about that MMSE is validated using its total score, but sub-scores are not. However, Choe YM et al. had evaluated the usefulness of subscores on the Mini-Mental State Examination (MMSE) for predicting the progression of Alzheimer’s disease (AD) dementia in individuals with mild cognitive impairment (MCI). The found the MMSE memory, orientation, and construction subscores, which are simple and readily available clinical measures, could provide useful information to predict AD dementia progression in MCI individuals in practical clinical settings. (Neuropsychiatr Dis Treat. 2020 Jul 24;16:1767-1775.) Therefore, we are also very interested to want to know the effect of each air pollutants on its five subgroups of cognitive functioning (G1-G5).
  3. Although the role of the health variables is controlled, it might be studied in order to confirm a complementary role in cognitive performance. It is hard to imagine an exclusive direct effect of the air pollution in cognition, but a more complex mediation effect.
  4. Ans: Thank you for your question. We tried to show the role of health variables in cognitive function. Therefore, we add Table 2 to show univariable analysis for total MMSE score using linear regression analysis.
  • Table 2 shows the determinants of total MMSE score in the study participants using multivariable linear regression analysis. Old age (unstandardized coefficient β, -0.006; p = 0.009), female (unstandardized coefficient β, 0.534; p = 0.002), low education years (unstandardized coefficient β, 0.950; p = 0.039), high BMI (unstandardized coefficient β, -0.092; p = 0.001), low O3 (unstandardized coefficient β, 0.047; p < 0.029), and high SO2 (unstandardized coefficient β, -0.145; p = 0.022) were associated with low total MMSE score. (Page 8, Line 11-17)
  •  

Other comments:

  1. In the abstract, the sentence "Our results showed that exposure to different air pollutants may lead to general cognitive decline and impairment of specific domains of cognitive functioning, except for ozone" it is not clear.  
  2. Ans: Thank you for your suggestion. We have revised to “Our results showed that exposure to different air pollutants may lead to general cognitive decline and impairment of specific domains of cognitive functioning, and O3 may be a protective factor.” (Page 3, Line 17-19)
  3. Introduction must be longer, with a more in deep review of the potential effects of air pollution in its different component on cognitive functioning (as it is developed in the Discussion, but with a bigger emphasis in its theoretical basis)
  4. Ans: Thank you for your suggestion. We have added more in the Introduction Paragraph 2.
  • The experimental researches have demonstrated that PM exposure can lead to diffuse accumulation of cerebral beta amyloid plaques, and hyperphosphorylated tau pre-tangles, and peripheral systemic inflammations, which further result in activation of microglial and astrocyte in central nervous system (CNS) and neuroinflammation has been implicated as an important pathway in neurodegenerative disease [5]. O3 per se is a strong oxidative pollutant, which elicits neurological injury by induction of release of free radicals, activation the generation of inflammatory cytokines and compromising the integrity of blood-brain barrier. Despite most evidence showed its detrimental influence on cognition, a few epidemiological studies revealed protective effect at low concentrations of O3 exposure [6]. (Page 4, Line 20 to Page 5, Line 3)
  •  
  1. How cognitive assessment and assessment of air pollutants are temporarily linked? Ans: We believe that the relationship between air pollutants and cognitive decline takes a period of time, not a short period of time can be caused. However, this study is a cross-sectional study, and therefore we could not examine the effects of long-term exposure to these air pollutants on cognitive functioning. Follow-up studies are needed to confirm our results. We had added this issue in the Limitation.
  • First, we believe that the relationship between air pollutants and cognitive decline takes a period of time, not a short period of time can be caused. it is a cross-sectional study, and therefore we could not examine the effects of long-term exposure to these air pollutants on cognitive functioning. Follow-up studies are needed to confirm our results. (Page 14, Line 25 to Page 15, Line 3)

Reviewer 3 Report

The enclosed manuscript describes a study of association between air pollution and poor cognitive function. The purpose of the study was to investigate associations between the most well-known atmospheric air pollutants and cognitive impairment.

Materials and methods

Evaluation of cognitive function

I do not see when and how cognitive function was assessed. Authors only described MMSE. Please include when and how cognition function was evaluated would be interesting to see.

In addition, this study would be more compelling if authors would be able to determine the effect of separate air pollutant on cognition.

Author Response

Reviewer 3

The enclosed manuscript describes a study of association between air pollution and poor cognitive function. The purpose of the study was to investigate associations between the most well-known atmospheric air pollutants and cognitive impairment.

Materials and methods

Evaluation of cognitive function

  1. I do not see when and how cognitive function was assessed. Authors only described MMSE. Please include when and how cognition function was evaluated would be interesting to see.
  2. Ans: Thank you for your question. We have added some sentences to show when and how cognitive function was assessed in Methods.
  • The participants completed MMSE questionnaires in face-to-face interviews with TWB researchers on the day of the enrollment. (Page 6, Line 16-18)
  •  
  1. In addition, this study would be more compelling if authors would be able to determine the effect of separate air pollutant on cognition.
  2. Ans: Thank you for your suggestion. We add Table 2 to show univariable analysis to show each air pollutants for total MMSE score using linear regression analysis.
  • Table 2 shows the determinants of total MMSE score in the study participants using multivariable linear regression analysis. Old age (unstandardized coefficient β, -0.006; p = 0.009), female (unstandardized coefficient β, 0.534; p = 0.002), low education years (unstandardized coefficient β, 0.950; p = 0.039), high BMI (unstandardized coefficient β, -0.092; p = 0.001), low O3 (unstandardized coefficient β, 0.047; p < 0.029), and high SO2 (unstandardized coefficient β, -0.145; p = 0.022) were associated with low total MMSE score. (Page 8, Line 11-17)

Round 2

Reviewer 1 Report

Thanks for your improvement!

Author Response

Reviewer 1

Thanks for your improvement!

Ans: Thank you for your review to make our article improved.

Reviewer 2 Report

Thank you to the authors for their review. Regarding my first comment, explanation about why they are using subscores must be added to the text. Complementarily, limitations of using a cognitive screening test and, in concrete, in using MMSE test must be addressed in the Discussion as limitations. The Introduction can be further extended. 

Author Response

Reviewer 2

Thank you to the authors for their review. Regarding my first comment, explanation about why they are using subscores must be added to the text. Complementarily, limitations of using a cognitive screening test and, in concrete, in using MMSE test must be addressed in the Discussion as limitations. The Introduction can be further extended. 

Ans: Thank you for your comments. We have added the explanation about why we use subscores in the Introduction. Besides, we added the limitations of using a cognitive screening test (MMSE).

  • Choe YM et al. had evaluated the usefulness of subscores on the MMSE for predicting the progression of Alzheimer’s disease dementia in individuals with mild cognitive impairment. The found the MMSE memory, orientation, and construction subscores, which are simple and readily available clinical measures, could provide useful information to predict Alzheimer’s disease dementia progression in mild cognitive impairment individuals in practical clinical settings [9]. Therefore, we are also very interested to want to know the effect of each air pollutants on its five subgroups of cognitive functioning. (Page 5, Line 12-19)
  • MMSE is a screening tool, and it only provides information about cognitive status. As a screening test, it is validated using its total score. Although different items use different cognitive functions, and some of them are summed in sub-scores, they are not actually sub-scales without psychometric information. (Page 15, Line 11-15)

Reviewer 3 Report

The changes made to the manuscript have improved the paper

Author Response

Reviewer 3

The changes made to the manuscript have improved the paper

Ans: Thank you for your review to make our article improved.